# Identification of Sildenafil Compound in Selected Drugs Using X-ray Study and Thermal Analysis

**DOI:** 10.3390/molecules28062632

**Published:** 2023-03-14

**Authors:** Izabela Jendrzejewska, Tomasz Goryczka, Ewa Pietrasik, Joanna Klimontko, Josef Jampilek

**Affiliations:** 1Institute of Chemistry, University of Silesia, Szkolna 9, 40007 Katowice, Poland; 2Institute of Materials Science, University of Silesia, Bankowa 12, 40007 Katowice, Poland; 3Institute of Physics, University of Silesia, Uniwersytecka 4, 40007 Katowice, Poland; 4Department of Analytical Chemistry, Faculty of Natural Sciences, Comenius University, Ilkovicova 6, 842 15 Bratislava, Slovakia; josef.jampilek@gmail.com

**Keywords:** sildenafil, X-ray study, quality phase analysis, thermal measurements

## Abstract

Twelve drugs containing sildenafil compounds (sildenafil citrate and sildenafil base) were examined using X-ray studies and thermal analysis. According to the manufacturer’s information, the presence of sildenafil was confirmed in all investigated drugs. The positions of diffraction lines (value of 2*θ* angle) agree with the patterns presented in the ICDD database, Release 2018 (ICDD—International Centre of Diffraction Data). The difference expresses the agreement in the position of the diffraction line between the tested substance and the standard. A good agreement is when this difference is less than 0.2°. The values of interplanar distances *d_hkl_* are also compatible with the ICDD database. It indicated that the drug examined was genuine. Because all drugs are mixtures of different substances (API and excipients), the various diffraction line intensities were detected in all observed X-ray images for the tested drugs. The intensity of the diffraction line depends on many factors, like the amount of substance, coexisting phases, and mass absorption coefficient of the mixture. The thermal analysis confirmed the results obtained by the X-ray study. On DSC curves, the endothermic peaks for sildenafil compounds were observed. The determined melting points of sildenafil compounds corresponded to the values available in the literature. The results gathered by connecting two methods, X-ray study and thermal analysis, can help identify irregularities that may exist in pharmaceutical specimens, e.g., distinguishing genuine from counterfeit products, the presence of a correct polymorph, a lack of active substance, an inaccurate amount of the active substance, or excipients in the tested drug.

## 1. Introduction

Since 1985, Pfizer chemists have been working to develop new phosphodiesterase (PDE) inhibitors that degrade cyclic guanosine-3′,5′-monophosphate (cGMP) as potential drugs for cardiovascular diseases (hypertension and angina pectoris). Sildenafil (see Figure 1), a synthetic analogue of purine compounds, was first prepared by Bell, Brown, and Terrett in 1989 and soon became a lead compound after biological and pharmacokinetic tests [1,2]. Two years later, the first clinical trials of sildenafil for the treatment of angina pectoris were initiated within the Pfizer company. This study concluded that there are several side effects; among the most common, such as headache and muscle pain, spontaneous erection in men was surprisingly recorded. Since such a strong side effect had never been observed before and insignificant effects of sildenafil in the treatment of angina pectoris had been demonstrated in phase I clinical testing, Pfizer decided to reconsider the strategy, and the first phase II clinical trials of sildenafil have already focused directly on the treatment of erectile dysfunction (ED) [1].

On a side note, it is good to remember that more than half of men over 40 suffer from some form of ED, and around 50% of them go untreated. In many cases, ED is the first symptom of common or even serious diseases such as hypertension, diabetes, cardiovascular disease, nervous system disorders (multiple sclerosis, Parkinson’s disease), and others [3,4].

Based on the results of the studies, Pfizer applied to the FDA for registration of sildenafil under the trade name Viagra^®^ (containing sildenafil citrate) for the treatment of ED, and in 1998 this “New Drug Application” No. 20-8954 was approved [1,5,6]. After the launch of Viagra^®^ by the pharmaceutical company Pfizer Inc. (New York, NY, USA) [6], there was a wave of publicity and interest from both scientists and potential users themselves, which was to some extent caused by the atypical shape and colour of the tablet. However, it must be acknowledged that the medicament Viagra^®^ revolutionised the treatment of ED mainly due to a sophisticated marketing strategy.

The discovery of sildenafil and, above all, its “innovative” application for the treatment of ED meant an economic success for Pfizer, which generated enormous interest in this molecule among its competitors. In 2013, Viagra^®^ patent protection expired in Europe and Canada, prompting the expansion of generic equivalents into the market. Currently, the worldwide sale of generic Viagra^®^ is allowed (in the USA, Pfizer managed to defend the effectiveness of the patent and thus maintain exclusivity until April 2020) [7]. After its launch in 1998 in the USA and the countries of the European Union, it became the fastest-selling drug ever by 2012, which speaks volumes about the economic potential of this molecule [8,9]. However, the market is currently full of generic products containing sildenafil, and these products are also a grateful target for many drug counterfeiters [6,10].

The results of investigations of the original Viagra^®^ tablets and some counterfeit imitations of Viagra were described in [11].

This work presents research results on various drugs containing sildenafil as an active pharmaceutical ingredient (API). Our work is a continuation of our previous study, which focused on examination drugs with acetaminophen and acetylsalicylic acid as well as on various dietary supplements using a combination of the following methods: X-ray Powder Diffraction method (XRPD), Differential Scanning Calorimetry (DSC), and Thermogravimetry (TG) [12,13,14,15]. The main goal of this work was to use phase analysis to identify sildenafil. Moreover, we tried to present the attempt to determine whether the product was original. Nine Over-the-Counter (OTC) drugs were chosen for this research, and three were on prescription.

## 2. Results and Discussion

### 2.1. X-ray Analysis

Our investigations focused on the qualitative analysis of drugs containing sildenafil API. Generally, drugs’ phase composition comprises API(s) and excipients. Each drug is a mixture of various components. The examined drugs include the following sildenafil APIs: sildenafil citrate and sildenafil base. The X-ray diffraction images of both these compounds are presented in Figure 2. The images were prepared using X-ray data from the X-ray diffraction database ICDD PDF4 (ICDD—International Centre of Diffraction Data, PDF—Powder Diffraction File) [16].

For sildenafil citrate, the diffraction line with the 100% relative intensity exists at 2*θ* = 14.4612° (112), whereas, for sildenafil base, the diffraction line with the 100% relative intensity exists at 2*θ* = 5.1545° (020). The qualitative X-ray analysis makes it possible to identify crystalline phases in the investigated samples. Each polycrystalline compound possesses a typical X-ray image consisting of diffraction lines. Each diffraction line has a specific position and intensity. The diffraction lines are formed even when the size of crystallites is in the order of a dozen or so unit cells in the direction of each crystallographic axis. It is, therefore, possible to identify substances with submicroscopic grain sizes. The particular place is expressed by the value of the 2*θ* angle at which the diffraction phenomenon occurs. The intensity of the line defines the number of counts.

It is worth highlighting that each polycrystalline phase retains its characteristic arrangement of diffraction lines in the mixture. For this reason, the substance’s X-ray image is called “fingerprints”. This procedure compares the experimental diffraction data (2*θ* diffraction angles, *d_hkl_* interplanar distances) with the data from the ICDD database [17]. Values of *d_hkl_* interplanar distances were calculated based on the Bragg–Wulff equation. The most important thing is to compare the 2*θ* angles because the shift of the line by more than 0.2° at a given diffraction angle of 2*θ* will indicate the presence of a different crystal structure. Such a phenomenon in an examined sample can be considered suspicious [18,19]. The absence of diffraction lines does not mean that a given phase is absent. Its amount may be below the X-ray detection limit (“sensitivity of the method”), i.e., below the lower percentage of the substance at which its diffraction pattern is not registered [12,13].

The results of the X-ray study of selected drugs with sildenafil are shown in Figure 3, Figure 4 and Figure 5 and Table 1 and Table 2. The value *J*/*J_max_* presents the relative intensity of the diffraction line, expressed as the ratio of the intensity of a given diffraction line to the intensity of the strongest diffraction line visible in the diffraction pattern.

#### 2.1.1. X-ray Study for Drugs with Sildenafil Citrate

X-ray diffraction patterns for drugs containing sildenafil citrate API are shown in Figure 3 and Figure 4.

Figure 3 and Figure 4 show that the diffraction lines derived from sildenafil citrate are visible. The diffraction lines align with the ICDD database pattern (PDF 00–052–2420). We considered six of the strongest diffraction lines visible on the diffraction pattern to compare the value of 2*θ* angle and calculations of interplanar distances *d_hkl_*.

Values |Δ2θ| are below 0.2°, which indicates the presence of sildenafil citrate in the examined drugs. Furthermore, the good agreement indicates that the sildenafil citrate is in proper crystallographic form (i.e., the same structural parameters). Additionally, the values of interplanar distances *d_hkl_* are consistent with the data presented in the ICDD database. We observed only the inconsistent values in the diffraction line’s intensities. The strongest line of examined drugs is placed at another 2*θ* angle compared to the pattern from the ICDD database (Table 1). The intensity of diffraction lines depends on their amount in the mixture, the crystal structure, the nature of the phases coexisting with it, and the mass absorption coefficient. The observed change in diffraction line intensity can indicate that in these drugs, the preferred grain orientation occurs at different values 2*θ* angle.

In the diffraction pattern for Sildena^®^ and Ernafil^®^, the strongest diffraction lines derived from Mg stearate and lactose monohydrate were detected. The line with the highest intensity belongs to Mg stearate, probably because a preferred orientation of the grains appeared in Mg stearate, significantly affecting the line’s intensity. Viagra^®^ also contains Mg stearate and lactose monohydrate. Still, their diffraction lines are overlapped by the diffractograms of amorphous components, which indicates that they are present in a high proportion in the composition of Viagra^®^.

The presence of Mg stearate and lactose monohydrate is also confirmed for the drugs presented in Figure 4. Although the convolution of three diffraction lines deriving from Mg stearate, lactose monohydrate, and sildenafil citrate is observed. This phenomenon is a result of very similar values of 2*θ* angle for the diffraction lines (lactose monohydrate—19.6679°, Mg stearate—19.7876°, and sildenafil citrate—19.8997°). The values of 2*θ* angle for Mg stearate and lactose monohydrate correspond to the positions of the strongest diffraction lines.

#### 2.1.2. X-ray Study for Drugs with Sildenafil Base

The results of the X-ray study for drugs with a sildenafil base are shown in Figure 5.

For tested drugs containing sildenafil base, the diffraction lines’ positions agree with the sildenafil base data (PDF Card 00–052–2006, Figure 2b), which confirms the presence of this compound in these drugs. The comparison of the experimental values of 2*θ* angle with the values presented in the ICDD database pointed out that the absolute value of the difference between these values is less than 0.2° (Table 2). For these drugs, the diffraction lines, typical for Mg stearate and lactose monohydrate are visible (Figure 5). The line of Mg stearate is the strongest, which indicates that the preferred grain orientation is detectable in the same phenomenon mentioned earlier.

Similarly, to the data for drugs with sildenafil citrate, the calculations done for drugs with sildenafil citrate indicate that the sildenafil base given in the examined drugs is in good crystal form.

### 2.2. Thermal Analysis

#### 2.2.1. Drugs with Sildenafil Citrate

Figure 6, Figure 7 and Figure 8 present the results of thermal analysis for drugs containing sildenafil citrate. Figure 6 shows the DSC (DSC—Differential Scanning Calorimetry) and TG/DTG (TG—Thermogravimetry, DTG—Derivative of Thermogravimetry) curves for drugs with sildenafil citrate that are available by prescription.

According to the literature data, the DSC curve for Viagra^®^ shows that the first endothermic peak appears at a temperature of 189 °C [20,21]. It is worth highlighting that, based on much of the literature data, the melting point of pure sildenafil citrate is in the range 182–196 °C [22,23]. Whereas, in Ref. [24], the sharp endothermic peak indicating the melting point of sildenafil citrate is reported at 203.7 °C. The thermoanalytical DSC and TG curves of Viagra^®^ are depicted in Figure 6. A small mass loss (about 1.5%) is observed on the TG curve, which is connected with the dehydration of sildenafil citrate. The next mass loss (about 6%) is observed in the temperature range between 185 °C and 205 °C and is connected with the citric acid evaporation [24], as well as with the process of melting of Viagra^®^. The endothermic peak, visible at 189 °C (Figure 6), indicates the phase transition between the solid state and liquid of Viagra^®^.

In the temperature range between 280 °C and 340 °C, a significant mass loss is observed (about 50%). The second endothermic peak appears at 303 °C. The place of this peak is consistent with the peak on DTG curves. At this temperature, the incineration of Viagra^®^ takes place. The flash point’s determined value is lower than that of pure sildenafil citrate (360 °C) [25,26]. In a mixture of substances, the endothermic peaks may be shifted towards lower temperatures.

DSC curves for Sildena^®^ and Ernafil^®^ show the first endothermic peak at 148 °C. It indicates the melting point of Mg stearate (*m_p_* 130–150 °C [27,28]), which is given in the composition of these drugs. It should be noted that the melting point value for Mg stearate is various and depends on the source, but Merck gives the value 140 °C [29]. Many sources present a temperature of 88 °C or 200 °C as a melting temperature [30,31].

The second endothermic peak observed on the DSC curves of examined drugs is a combination of two peaks: the first part indicates the melting point of sildenafil citrate (*m_p_* 184–196 °C [22,23]) at temperatures of 188 °C and 183 °C for Sildena^®^ and Ernafil^®^, respectively. The second part of the peak indicates the melting point of lactose monohydrate (*m_p_* 202–215 °C [32,33]) at temperatures of 202 °C and 199 °C for Sildena^®^ and Ernafil^®^, respectively. The results of DSC/TG measurements correspond to those obtained by X-ray analysis. For both drugs, the decomposition and mass loss take place above 330 °C (Figure 6a,b). The results of DSC/TG measurements correspond to those obtained by X-ray analysis.

Figure 7 and Figure 8 show the DSC and TG/DTG curves for OTC drugs containing sildenafil citrate.

The shape of the DSC and TG/DTG curves for OTC drugs with sildenafil citrate is very similar and agrees with the Viagra DSC/TG curves and the literature data. The melting point for tested drugs has almost the same value (189–191 °C), while the decomposition temperature is higher than that of pure sildenafil citrate and Viagra^®^. It may be connected with the presence of different excipients in the examined drugs.

The thermal parameters such as onset, offset, peak maximum, and mass loss for drugs containing sildenafil citrate are presented in Table 3.

#### 2.2.2. Drugs with Sildenafil Base

The shape of the DSC and TG/DTG curves for drugs with a sildenafil base is very similar (Figure 9 and Figure 10). The first endothermic peak, visible at a temperature range of 140–147 °C, confirms the presence of Mg stearate in the composition of the tested drugs. In this range of temperature, a slight loss of weight is observed. It means that only a small part of the drug melted. The diffraction lines derived from Mg stearate are noticeable in diffraction patterns (Figure 5).

The small second endothermic peaks visible in Figure 9 and Figure 10 are at temperatures of 181–183 °C. At this temperature, no loss of mass is observed. It indicates that at this temperature, the melting process of the sildenafil base is observed. The third endothermic peak, visible in the range of temperatures 198–202 °C, indicates the melting point of lactose monohydrate, which is included. The DSC/TG results are in good agreement with the X-ray study. Together with this peak, the substantial loss mass is observed on the TG/DTG curves. The thermal parameters for this group of drugs are presented in Table 4.

## 3. Materials and Methods

The drugs containing sildenafil compounds were examined using the following methods: X-ray Powder Diffraction (XRPD), Thermogravimetry (TG), and Differential Scanning Calorimetry (DSC).

### 3.1. Materials

Twelve drugs containing sildenafil compounds were selected for the examination. Nine drugs belong to the Over-the-Counter (OTC) group of drugs. The rest of the drugs are prescription drugs: Viagra^®^, Ernafil^®^, and Sildena^®^. All the examined drugs are listed in Table 5, where the details containing information about the manufacturers and the amount of sildenafil compound per tablet are presented.

### 3.2. Methods

#### 3.2.1. X-ray Analysis of Samples of Selected Drugs Containing Sildenafil

The tablets were crushed thoroughly in an agate mortar to obtain an excellent, homogeneous powder. Next, such prepared samples were examined using polycrystalline diffractometers: D5000 (Siemens, Munich, Germany) and PW3050 (X’Pert Philips, Malvern Panalytical, Malvern, UK), using Bragg–Brentano geometry and CuKα radiation. The time of analysis for every sample was approx. 2 h. All technical details about X-ray analysis are presented in [13,14,15]. All diffraction images are shown for a *2θ* angular range of 5–30°, as the strongest diffraction lines are visible at low angles. To prepare a comparative analysis between the experimental data and the standards, we applied diffractometric data from the ICDD PDF4 database (Release 2018) [16]. For some drugs, the diffraction lines originating from excipients were identified. The numbers of the PDF cards used for diffraction line analysis are presented in Table 6.

#### 3.2.2. Thermal Analysis

The small amount of the tested drug (below 10 mg) was located in an aluminum crucible with a lid and heated to a temperature of 450 °C under an argon atmosphere, using a heating rate of 5 °C/min. The LabsysEvo (Setaram Inc., Cranbury, NJ, USA) apparatus was used for the thermal analysis. For every sample, the measurement was performed under the same conditions.

## 4. Conclusions

Studies on identifying sildenafil APIs in selected drugs were carried out using X-ray and thermal analyses. The tests were done for three prescription drugs and nine commonly available drugs in pharmacies. The X-ray analysis compared diffraction line positions (*2θ* angles) for the tested samples, their intensities, and the interplanar distance *d_hkl_* with data from the ICDD PDF4 database. Based on the performed measurements, it was found that the tested samples contain a sildenafil compound, as declared by the manufacturer. The difference between the values of the experimental *2θ* angles and the data from a database is less than 0.2°, and the diffraction patterns are consistent with the images in the ICDD database. It indicates the use of a substance with the same structural parameters or substances with the same crystal form. Differences in the intensities of diffraction lines derived from sildenafil compounds were observed compared to data from a database. It should be highlighted that each drug is a mixture of crystalline and amorphous substances (API(s) and excipients). The intensity of the diffraction line depends not only on its amount in the examined mixture but also on the crystal structure, coexisting phases, and mass absorption coefficient of the mixture. Data presented in X-ray databases concern pure substances.

The DSC and TG curves analysis confirmed the presence of sildenafil API in the examined drugs. For all medicines, an endothermic peak (at a temperature range of 181–188 °C) was visible on the DSC curve, indicating the presence of sildenafil. Endothermic peaks originating from Mg stearate and lactose monohydrate were observed for some of the medicines. This phenomenon is in excellent agreement with the results of X-ray studies, where diffraction lines from these auxiliary substances are visible in the diffraction pattern.

Studies using a combination of two techniques, thermal analysis, and X-ray investigations, can identify active substances in drugs and distinguish genuine drugs from counterfeit, e.g., by checking the presence of the correct active substance or determining the presence of its proper polymorph. Thermal measurements allow us to determine the temperature limits below which the analysed substances can be processed without changing their physicochemical properties. The results presented in this work may be helpful as a guideline for testing the stability of various drugs and may be used to detect incompatibilities in the composition of medicines. If the X-ray study and thermal measurements provide questionable data, other methods, e.g., IR, UV-Vis, XRF, chromatographic, or microscopic, should be used [34,35,36].

## Figures and Tables

**Figure 1 molecules-28-02632-f001:**
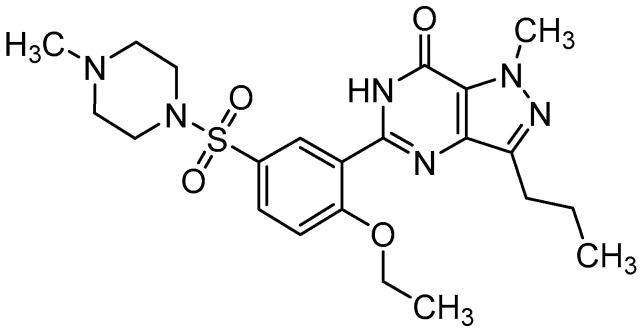
Structure of sildenafil (5-{2-ethoxy-5-[(4-methylpiperazin-1-yl)sulfonyl]phenyl}-1-methyl- 3-propyl-1,6-dihydro-7*H*-pyrazolo [4,3-*d*]pyrimidin-7-one), CAS 139755-83-2, ATC code G04BE03.

**Figure 2 molecules-28-02632-f002:**
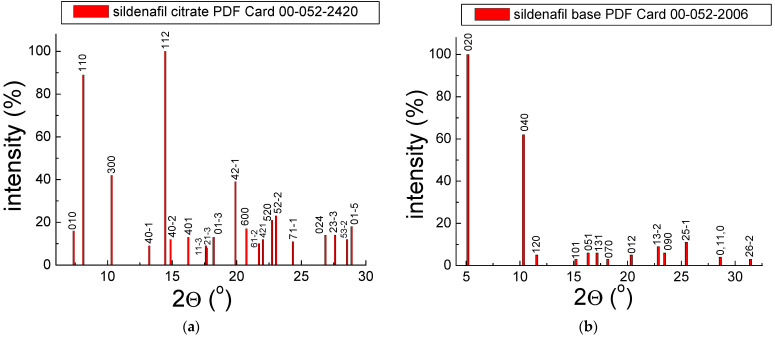
X-ray diffraction images for sildenafil citrate (**a**) and sildenafil base (**b**).

**Figure 3 molecules-28-02632-f003:**
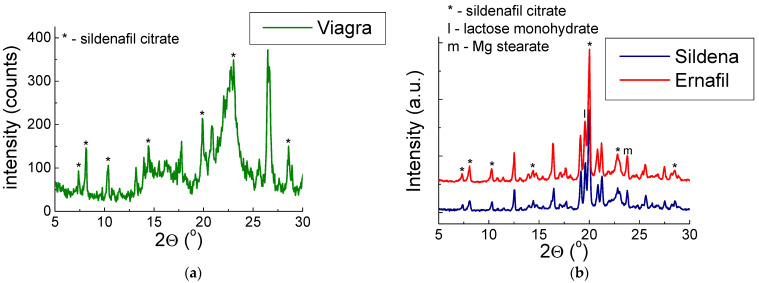
X-ray pattern for drugs containing sildenafil citrate: (**a**) Viagra^®^, (**b**) Sildena^®^, and Ernafil^®^. These drugs require a prescription.

**Figure 4 molecules-28-02632-f004:**
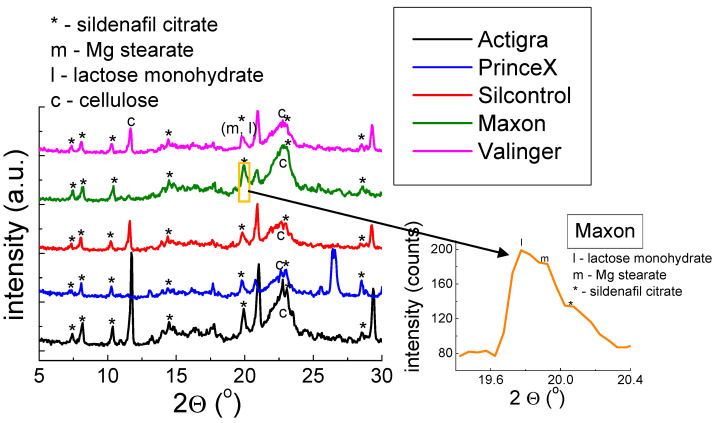
X-ray pattern for Over-the-Counter (OTC) drugs containing sildenafil citrate. Inset: the convolution of three diffraction lines of Mg stearate, lactose monohydrate, and sildenafil citrate.

**Figure 5 molecules-28-02632-f005:**
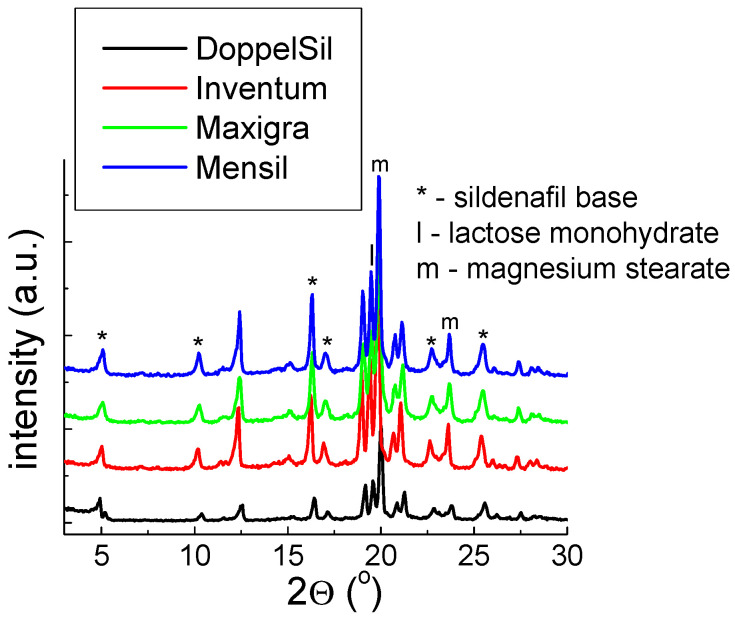
X-ray pattern for Over-the-Counter (OTC) drugs containing sildenafil base.

**Figure 6 molecules-28-02632-f006:**
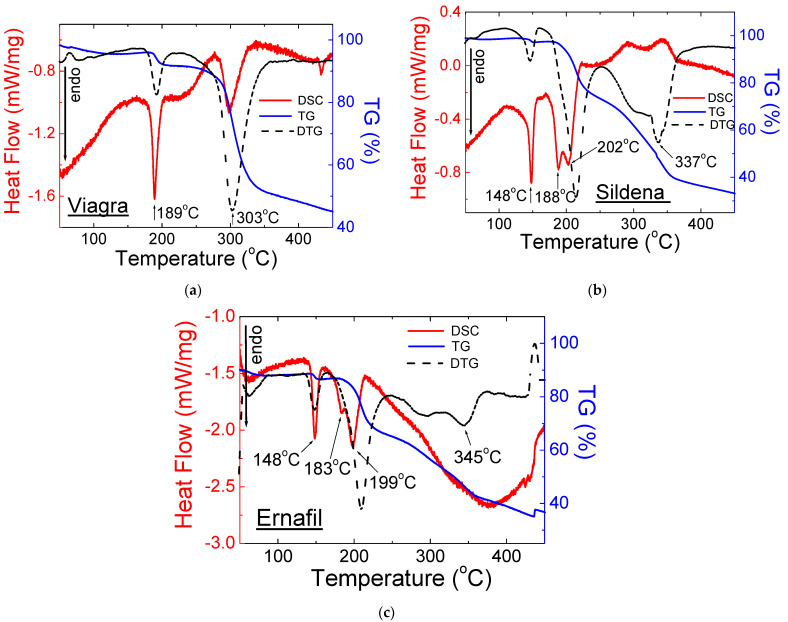
X-ray DSC and TG/DTG curves for drugs on prescription: (**a**) Viagra^®^, (**b**) Sildena^®^, and (**c**) Ernafil^®^.

**Figure 7 molecules-28-02632-f007:**
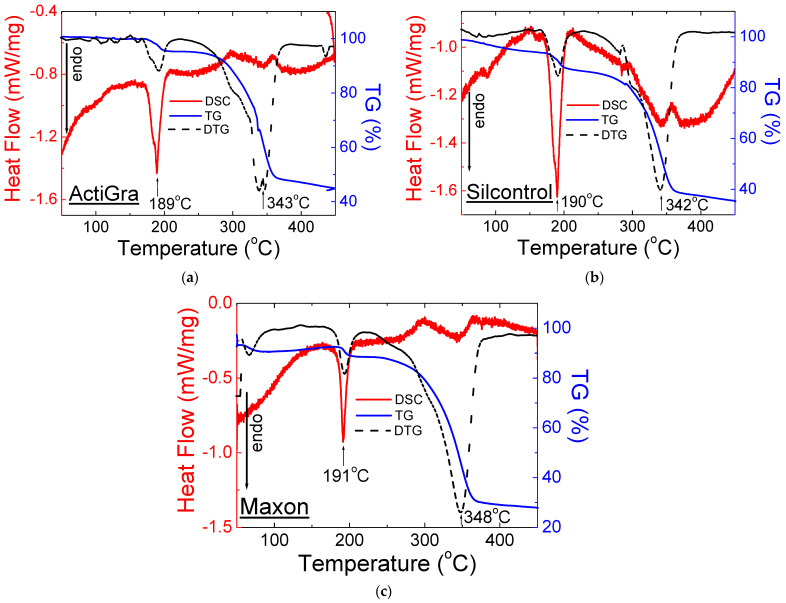
DSC and TG/DTG for Over-the-Counter (OTC) drugs with sildenafil citrate: (**a**) ActiGra^®^, (**b**) Silcontrol^®^, and (**c**) Maxon^®^.

**Figure 8 molecules-28-02632-f008:**
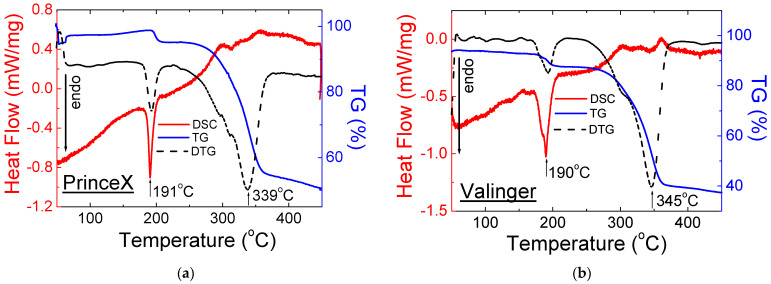
DSC and TG/DTG for Over-the-Counter (OTC) drugs with sildenafil citrate: (**a**) PrinceX^®^ and (**b**) Valinger^®^.

**Figure 9 molecules-28-02632-f009:**
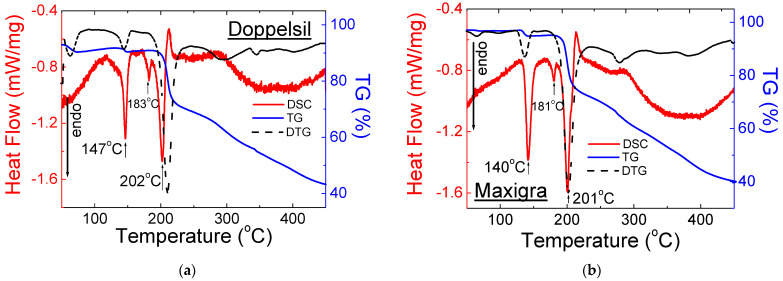
DSC and TG/DTG for Over-the-Counter (OTC) drugs with a sildenafil base: (**a**) Doppelsil^®^ and (**b**) Maxigra^®^.

**Figure 10 molecules-28-02632-f010:**
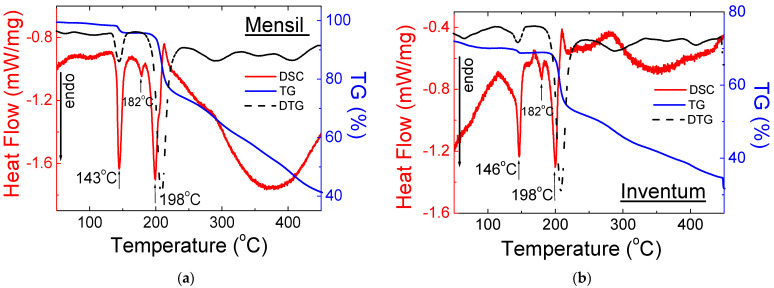
DSC and TG/DTG for Over-the-Counter (OTC) drugs with a sildenafil base: (**a**) Mensil^®^ and (**b**) Inventum™.

**Table 1 molecules-28-02632-t001:** Comparison of experimental and diffraction data from the ICDD database for drugs containing sildenafil citrate (C_28_H_38_N_6_O_11_S) compatible with the PDF card 00–052–2420.

No. of Diffraction Line	2*θ* (^o^) Exp.	2*θ* (^o^) ICDD	J/J_max_ (%) Exp.	J/J_max_ (%) ICDD	|Δ2*θ*| *	d_hkl_ (Å) Exp.	d_hkl_ (Å) CDD	hkl **
**Viagra^®^**								
1.	7.3981	7.3730	27	16	0.0251	11.94	11.98	010
2.	8.1870	8.1122	41	89	0.0748	10.79	10.89	110
3.	10.2926	10.3135	33	42	0.0209	8.59	8.57	300
4.	14.4514	14.4612	42	100	0.0098	6.12	6.12	112
5.	19.9320	19.8997	57	39	0.0323	4.45	4.46	42-1
6.	23.0929	23.0460	93	23	0.0469	3.85	3.86	52-2
**Ernafil^®^**								
1.	7.3615	7.3730	10	16	0.0115	12.00	11.98	010
2.	8.0668	8.1122	15	89	0.0454	10.95	10.89	110
3.	10.3448	10.3135	13	42	0.0313	8.54	8.57	300
4.	14.3992	14.4612	11	100	0.0620	6.15	6.12	112
5.	20.0366	19.8997	100	39	0.1369	4.43	4.46	42-1
6.	22.9624	23.0460	25	23	0.0836	3.87	3.86	52-2
**Sildena^®^**								
1.	7.3824	7.3730	11	16	0.0094	11.96	11.98	010
2.	8.0564	8.1122	14	89	0.0558	10.97	10.89	110
3.	10.2508	10.3135	13	42	0.0313	8.62	8.57	300
4.	14.4671	14.4612	14	100	0.0059	6.12	6.12	112
5.	19.9739	19.8997	100	39	0.0742	4.44	4.46	42-1
6.	23.0407	23.0460	63	23	0.0053	3.86	3.86	52-2
**Valinger^®^**								
1.	7.3981	7.3730	25	16	0.0251	11.94	11.98	010
2.	8.0825	8.1122	32	89	0.0297	10.93	10.89	110
3.	10.2926	10.3135	28	42	0.0209	8.59	8.57	300
4.	14.4514	14.4612	38	100	0.0098	6.12	6.12	112
5.	19.8276	19.8997	43	39	0.0721	4.48	4.46	42-1
6.	23.2341	23.0460	53	23	0.1881	3.83	3.86	52-2
**ActiGra^®^**								
1.	7.3992	7.3730	20	16	0.0262	11.94	11.98	010
2.	8.2111	8.1122	29	89	0.0989	10.76	10.89	110
3.	10.3469	10.3135	27	42	0.0361	8.54	8.57	300
4.	14.4723	14.4612	32	100	0.0111	6.11	6.12	112
5.	19.9289	19.8997	45	39	0.0292	4.45	4.46	42-1
6.	23.0930	23.0460	64	23	0.0470	3.85	3.86	52-2
**Silcontrol^®^**								
1.	7.3824	7.3730	23	16	0.0094	11.96	11.98	010
2.	8.0564	8.1122	31	89	0.0558	10.96	10.89	110
3.	10.3083	10.3135	25	42	0.0052	8.57	8.57	300
4.	14.4096	14.4612	33	100	0.0516	6.14	6.12	112
5.	19.8641	19.8997	42	39	0.0356	4.47	4.46	42-1
6.	23.0407	23.0460	63	23	0.0053	3.86	3.86	52-2
**Maxon^®^**								
1.	7.4922	7.3730	23	16	0.1192	11.79	11.98	010
2.	8.1714	8.1122	31	89	0.0592	10.81	10.89	110
3.	10.3605	10.3135	31	42	0.0470	8.53	8.57	300
4.	14.5246	14.4612	43	100	0.0634	6.09	6.12	112
5.	19.9774	19.8997	66	39	0.0777	4.44	4.46	42-1
6.	23.0930	23.0460	96	23	0.0470	3.85	3.86	52-2
**PrinceX^®^**								
1.	7.3458	7.3730	25	16	0.0270	12.02	11.98	010
2.	8.0825	8.1122	38	89	0.0297	10.93	10.89	110
3.	10.3448	10.3135	33	42	0.0313	8.54	8.57	300
4.	14.3992	14.4612	28	100	0.0620	6.15	6.12	112
5.	19.8798	19.8997	40	39	0.0199	4.46	4.46	42-1
6.	23.0930	23.0460	62	23	0.0470	3.85	3.86	52-2

* absolute value of the difference between the experimental value of 2*θ* and the value of 2*θ* from the ICDD database; ** Miller’s indices of diffraction lines.

**Table 2 molecules-28-02632-t002:** Comparison of experimental and diffraction data from the ICDD database for drugs containing sildenafil base compatible with the PDF card 00–052–2006.

No. of Diffraction Line	2*θ* (^o^) Exp.	2*θ* (^o^) ICDD	J/J_max_ (%) Exp.	J/J_max_ (%) ICDD	|Δ2*θ*| *	d_hkl_ (Å) Exp.	d_hkl_ (Å) ICDD	hkl **
**DoppelSil^®^**								
1.	5.2089	5.1545	13	100	0.0544	16.95	17.13	020
2.	10.3866	10.3256	11	62	0.0610	8.51	8.56	040
3.	16.4159	16.3409	27	6	0.0750	5.40	5.42	051
4.	17.1525	17.1703	13	6	0.0178	5.17	5.16	131
5.	22.8631	22.8538	16	9	0.0093	3.89	3.88	090
6.	25.5695	25.4645	22	11	0.1050	3.48	3.49	25-1
**Maxigra^®^**								
1.	5.0236	5.1545	16	100	0.1309	17.58	17.13	020
2.	10.2202	10.3256	15	62	0.1054	8.65	8.56	040
3.	16.2660	16.3409	43	6	0.0749	5.44	5.42	051
4.	16.9891	17.1703	17	6	0.1812	5.21	5.16	131
5.	22.6831	22.8538	19	9	0.1707	3.92	3.88	090
6.	25.4501	25.4645	22	11	0.0144	3.50	3.49	25-1
**Mensil^®^**								
1.	5.0107	5.1545	17	100	0.1438	17.62	17.13	020
2.	10.1835	10.3256	16	62	0.1421	8.68	8.56	040
3.	16.2746	16.3409	44	6	0.0663	5.44	5.42	051
4.	16.9755	17.1703	15	6	0.1948	5.22	5.16	131
5.	22.6646	22.8538	18	9	0.1892	3.92	3.88	090
6.	25.4194	25.4645	20	11	0.0451	3.50	3.49	25-1
**Inventum™**								
1.	5.0236	5.1545	15	100	0.1309	17.58	17.13	020
2.	10.1865	10.3256	14	62	0.1391	8.68	8.56	040
3.	16.2379	16.3409	37	6	0.1030	5.45	5.42	051
4.	16.9797	17.1703	16	6	0.1906	5.22	5.16	131
5.	22.6694	22.8538	17	9	0.1844	3.92	3.88	090
6.	25.4164	25.4645	19	11	0.0481	3.50	3.49	25-1

* absolute value of the difference between the experimental value of 2*θ* and the value of 2*θ* from the ICDD database; ** Miller’s indices of diffraction lines.

**Table 3 molecules-28-02632-t003:** Thermal parameters obtained from DSC/TG analysis for selected drugs containing sildenafil citrate.

No.	Name of Drug	Weight Loss (%)	Onset (°C)	Offset (°C)	Peak Maximum (°C)	Peak Height (mW)	Peak Area (J)	Enthalpy (J/g)
**1**	Viagra^®^	55	185	194	189	3.75	0.52	72.6
			290	303	298	2.29	0.74	103
**2**	Ernafil^®^	65	144	153	148	3.10	0.25	52.3
			189	210	183, 199	3.00	0.67	140
**3**	Sildena^®^	68	144	151	148	4.48	0.54	75.1
			181	214	188, 202	4.47	1.50	209
**4**	Valinger^®^	63	177	197	190	3.87	0.60	97.5
			343	357	348	0.48	0.08	13.4
**5**	ActiGra^®^	55	185	196	189	3.36	0.51	93.3
			345	356	354	0.43	0.07	13.4
**6**	Silcontrol^®^	65	180	195	190	4.23	0.66	107
			328	334	334	0.77	0.28	44.4
**7**	Maxon^®^	72	187	196	191	3.02	0.31	66.4
			343	367	353	0.73	0.25	52.7
**8**	PrinceX^®^	49	187	197	191	5.05	0.69	103.1
			314	324	319	0.46	0.05	8.10

**Table 4 molecules-28-02632-t004:** Thermal parameters obtained from DSC/TG analysis for selected drugs containing a sildenafil base.

No.	Name of Drug	Weight Loss (%)	Onset (°C)	Offset (°C)	Peak Maximum (°C)	Peak Height (mW)	Peak Area (J)	Enthalpy (J/g)
**1**	DoppelSil^®^	57	142	150	147	2.50	0.28	68.0
			174	188	183	0.71	0.02	4.22
			195	207	202	2.96	0.23	55.0
**2**	Maxigra^®^	31	135	148	140	6.68	0.77	72.6
			179	183	181	1.35	0.08	7.70
			194	206	201	9.78	1.25	118
**3**	Mensil^®^	27	140	148	143	5.50	0.53	65.0
			180	186	182	0.88	0.05	5.73
			192	202	198	6.24	0.70	86.7
**4**	Inventum™	24	142	150	146	4.70	0.45	73.2
			180	186	182	0.88	0.04	5.73
			192	202	198	6.24	0.54	86.7

**Table 5 molecules-28-02632-t005:** Analysed drugs containing sildenafil compounds.

No.	Product Name (Manufacturer)	Sildenafil Compound Content in 1 Tablet (mg)	Mass (%) of Sildenafil Compound in 1 Tablet
**1**	Viagra^®^ (Pfizer)	100	15.7
**2**	Ernafil^®^ (Teva)	100	18.8
**3**	Sildena^®^ (Hemofarm)	100	16.2
**4**	Valinger^®^ (Orion Pharma)	25	16.1
**5**	Maxigra^®^ (Polpharma)	25	18.5
**6**	Mensil^®^ (Hasco-Lek S.A.)	25	18.8
**7**	DoppelSil^®^ (Doppelherz Pharma)	25	18.1
**8**	Inventum™ (Aflofarm Farmacja Polska)	25	18.9
**9**	Maxon^®^ (Adamed Pharma)	25	16.1
**10**	ActiGra^®^ (Biofarm)	25	16.6
**11**	Silcontrol^®^ (Zentiva)	25	16.1
**12**	PrinceX^®^ (Accord Healthcare)	25	16.6

**Table 6 molecules-28-02632-t006:** List of used ICDD PDF4 (release 2018) cards.

No.	Chemical Compound	Chemical Formula	No. PDF Card
**1**	sildenafil citrate	C_28_H_38_N_6_O_11_S	00–052–2420
**2**	sildenafil base	C_22_H_30_N_6_O_4_S	00–052–2006
**3**	lactose monohydrate	C_12_H_22_O_11_·H_2_O	00–001–0333
**4**	magnesium stearate	C_36_H_70_MgO_4_	00–005–0292
**5**	cellulose	C_6_H_10_O_5_	00–003–0223

## Data Availability

The data presented in this study are available upon request from the corresponding authors.

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
