# Peer review of "Identification of Sildenafil Compound in Selected Drugs Using X-ray Study and Thermal Analysis"

_molecules, 2023, doi:10.3390/molecules28062632_

Round 1

Reviewer 1 Report

1. For PXRD Reference Data (Section 2.1), it is necessary to mention main peak positions in text. It is hard to guess 2-theta angles from the figures.

2. The reference [24] and not cited source (DOI 10.1007/s10973-012-2292-8) clearly show that SC shows the evaporation of citric acid and fusion of the residual base. There is no need to cite the websites 22-23 as we have more clear scientific articles. The change of mass is significant for pure SC; thus, it is necessary to improve the discussion in Line 184 (not insignificant change of mass).

3. DSC figures must have arrows indicating exo (or endo) direction for the better readability of thermograms.

Author Response

We thank Reviewer 1 very much for carefully reading the paper and their constructive remarks. In order to take into account the latter, the paper has been revised. All changes are marked in yellow.

Review 1.

  1. For PXRD Reference Data (Section 2.1), it is necessary to mention main peak positions in text. It is hard to guess 2-theta angles from the figures.

Position information about the strongest diffraction lines for sildenafil citrate and sildenafil base has been added to the manuscript (lines 94, 95).

  1. The reference [24] and not cited source (DOI 10.1007/s10973-012-2292-8) clearly show that SC shows the evaporation of citric acid and fusion of the residual base. There is no need to cite the websites 22-23 as we have more clear scientific articles. The change of mass is significant for pure SC; thus, it is necessary to improve the discussion in Line 184 (not insignificant change of mass).

The websites have been removed, the more proper references have been added. The discussion about the changes of mass for Viagra has been improved and explained more clearly (lines 192 -201).

  1. DSC figures must have arrows indicating exo (or endo) direction for the better readability of thermograms.

The figures have been corrected according to Reviewer’s suggestion.

Reviewer 2 Report

The manuscript is focused on the identification of sildenafil in twelve different drugs by means of X-ray and thermal analytical studies. The comparison of the results obtained with the data from ICDD database had improved the validation of results by the authors. The manuscript is well-structured and well-written. However, I find that incorporation of a few following comments would help in enhancing the quality of the manuscript and help the readers to understand the concepts:

(1) The authors have used few terms like ICDD, J/Jmax, etc. without describing its expansion in the first place where the term has been mentioned. Kindly describe the term in the first place where it is mentioned.

(2) In the sentence in the line numbers 137-138, the authors have mentioned 'The high background indicates a large share of amorphous substances in the composition of Viagra®'. Can this sentence explained a bit clearly by removing the term'background'? I suppose that the word 'background' cannot serve as a suitable term for its use here.

(3) In the X-RD patterns shown in Fig. 4, pls do present the comparison data with representation of hkl values that corresponds to the presented diffractograms if possible.

Author Response

We thank Reviewer 2 very much for carefully reading the paper and their constructive remarks. In order to take into account the latter, the paper has been revised. All changes are marked in yellow.

Review 2.

The manuscript is focused on the identification of sildenafil in twelve different drugs by means of X-ray and thermal analytical studies. The comparison of the results obtained with the data from ICDD database had improved the validation of results by the authors. The manuscript is well-structured and well-written. However, I find that incorporation of a few following comments would help in enhancing the quality of the manuscript and help the readers to understand the concepts:

  • The authors have used few terms like ICDD, J/Jmax, etc. without describing its expansion in the first place where the term has been mentioned. Kindly describe the term in the first place where it is mentioned.

The abbreviations have been explained in the text. The explanation of J/Jmax is presented in lines 116 – 118.

  • In the sentence in the line numbers 137-138, the authors have mentioned 'The high background indicates a large share of amorphous substances in the composition of Viagra®'. Can this sentence explained a bit clearly by removing the term'background'? I suppose that the word 'background' cannot serve as a suitable term for its use here.

The sentence has been modified, the term "backgroud" has been removed.

  • In the X-RD patterns shown in Fig. 4, pls do present the comparison data with representation of hkl values that corresponds to the presented diffractograms if possible.

 The Miller’s indices (hkl) have been added to the standard X-ray images for sildenafil citrate and sildenafil base and to the tables with diffraction data (Fig. 2, Tables 1 and 2).  Miller’s indices (hkl) in Figure 4 would make the figure illegible.